# A Composite System Based upon Hydroxypropyl Cyclodextrins and Soft Hydrogel Contact Lenses for the Delivery of Therapeutic Doses of Econazole to the Cornea, In Vitro

**DOI:** 10.3390/pharmaceutics14081631

**Published:** 2022-08-04

**Authors:** Anepmete Wong, Melissa Fallon, Vildan Celiksoy, Salvatore Ferla, Carmine Varricchio, David Whitaker, Andrew J. Quantock, Charles M. Heard

**Affiliations:** 1School of Pharmacy and Pharmaceutical Sciences, Cardiff University, Wales CF10 3NB, UK; 2School of Optometry and Vision Sciences, Cardiff University, Wales CF24 4HQ, UK; 3School of Healthcare Sciences, Cardiff University, Wales CF14 4XN, UK

**Keywords:** cornea, fungal keratitis, econazole, contact lens, cyclodextrin, composite system

## Abstract

Fungal keratitis, a disease in which the cornea becomes inflamed due to an invasive fungal infection, remains difficult to treat due in part to limited choices of available treatments. Topical eye drops are first-line treatment, but can be ineffective as low levels of drug reach the target site due to precorneal losses and the impenetrability of the cornea. The aim of this study was to determine the corneal delivery of econazole using a novel topical enhancement approach using a composite delivery system based upon cyclodextrins and soft hydrogel contact lenses. Excess econazole nitrate was added to hydroxypropyl-α-cyclodextrin (HP-α-CD) and hydroxypropyl-β-cyclodextrin (HP-β-CD) solutions, and the solubility determined using HPLC. Proprietary soft hydrogel contact lenses were then impregnated with saturated solutions and applied to freshly enucleated porcine eyeballs. Econazole nitrate ‘eye drops’ at the same concentrations served as the control. After 6 h, the corneas were excised and drug-extracted, prior to quantification using HPLC. Molecular dynamic simulations were performed to examine econazole–HP-β-CD inclusion complexation and dissociation. The minimum inhibitory concentration (MIC) of econazole was determined against four fungal species associated with keratitis, and these data were then related to the amount of drug delivered to the cornea, using an average corneal volume of 0.19 mL. The solubility of econazole increased greatly in the presence of HP-β-CD and more so with HP-α-CD (*p* < 0.001), with ratios >> 2. Hydrogel contact lenses delivered ×2.8 more drug across the corneas in comparison to eye drops alone, and ×5 more drug delivered to the cornea when cyclodextrin was present. Molecular graphics demonstrated dynamic econazole release, which would create transient enhanced drug concentration at the cornea surface. The solution-only drops achieved the least satisfactory result, producing sub-MIC levels with factors of ×0.81 for both *Fusarium semitectum* and *Fusarium solani* and ×0.40 for both *Scolecobasidium tshawytschae* and *Bipolaris hawaiiensis*. All other treatments delivered econazole at > MIC for all four fungal species. The efficacies of the delivery platforms evaluated were ranked: HP-α-CD contact lens > HP-β-CD contact lens > contact lens = HP-α-CD drops > HP-β-CD drops > solution-only drops. In summary, the results in this study have demonstrated that a composite drug delivery system based upon econazole–HP-β-CD inclusion complexes loaded into contact lenses can achieve significantly greater corneal drug delivery with the potential for improved clinical responses.

## 1. Introduction

Fungal keratitis is the inflammation of the cornea caused by the invasion of fungi into the stroma and, although not a major problem in the Western world, cases of fungal keratitis are difficult to treat in developing countries, often due to delayed diagnosis [1]. Eye trauma, for example, incorrect contact lens usage, foreign bodies and prior corneal surgery, can result in defects in the epithelium, providing opportunistic microorganisms including fungi access into the corneal stroma, where they can proliferate, leading to host inflammatory reactions, vision loss and nerve damage [2,3]. Fungal keratitis presents in a similar manner to other types of microbial keratitis, and the problem is exacerbated by delays in diagnosing fungal keratitis and prompt administration of antifungal drugs and, once established, can be destructive and difficult to treat compared to bacterial keratitis [4]. Symptoms include severe pain, redness, blurred vision, photophobia, excessive tearing or discharge; later-stage infections include corneal ulceration. The major fungal groups that are responsible for fungal infection are filamentous (*Fusarium*, *Aspergillus* or *Candida*) and yeast fungi and initially present as feathery growths (Figure 1).

In terms of treatment, antifungal agents available are limited, and typically have poor aqueous solubility, which gives rise to issues in the topical formulations and poor corneal penetration of drugs. Natamycin eye drops are a typical first-line topical treatment in many countries; however, the drug possesses poor penetration into the cornea, and, as the extent of infection can vary widely, duration of treatment can take from a few weeks to months. A practice performed in some areas is the debridement of the corneal epithelium prior to application, in order to increase drug penetration rates; however, this is a highly invasive and risky treatment, requiring hospitalisation for several weeks.

Ultimately, surgical intervention with a penetrating or anterior lamellar keratoplasty, including corneal transplant, may be required if drug treatment fails to clear the infection [2,4]—these are also very invasive with a high risk of graft rejection [5]. Therefore, an alternative approach is needed to provide a safer and more effective fungal keratitis treatment. Econazole nitrate, a drug of the imidazole family, has broad-spectrum activity against dermatophytes, *Candida* species and other pathogenic yeasts and some Gram-positive bacteria. It has been proposed as an alternative treatment for fungal keratitis in several recent works such as a trial reported by Prajna et al. [6], which found no significant differences between 2% econazole and 5% natamycin for the management of fungal keratitis. Topical creams containing econazole are available, e.g., for athlete’s foot, but as far as we are aware there are no products currently indicated for ocular infections.

Topical eye drop instillation is the mainstay of ocular drug delivery and is generally well-tolerated by patients as it is non-invasive, easy to self-administer and convenient. However, precorneal factors such as blinking of the eye, tear production and nasolacrimal drainage [7] typically combine to result in low ocular bioavailability, i.e., <5% of the instilled dose reaching the deeper ocular tissues [8], with much of the rest being swallowed and entering the systemic circulation. In addition, various layers of the cornea present different environmental polarities for drug permeation. For example, the stroma, which comprises 90% of the corneal thickness, is a hydrophilic layer, and this poses a challenge for the permeation of lipophilic drugs. Increasing the amount of drug in the applied dose can generally result in improved bioavailability and therapeutic outcomes. Econazole-loaded hydrogels have been evaluated recently, although no modified permeability of econazole was found [9]. Cyclodextrins are cyclic oligosaccharides consisting of 6 (α), 7 (β) and 8 (γ) glucose units in a ring with a toroidal structure. They are hydrophobic on the interior, allowing inclusion complexation with guest molecules (i.e., drugs), whilst the exterior is relatively hydrophilic, especially in derivatives such as hydroxypropyl-α-cyclodextrin (HP-α-CD) and hydroxypropyl-β-cyclodextrin (HP-β-CD). Such cyclodextrins are thus able to increase the aqueous solubility of hydrophobic drugs [10,11] and enhance ocular drug delivery by transporting the lipophilic drug molecules through the aqueous tear film to the mucin layer, thereby increasing drug availability at the lipophilic corneal surface [12]. The use of cyclodextrins to increase the aqueous solubility of econazole is not a new concept [9,13,14]; however, the problem of nasolacrimal drainage remains, along with the issues of unwanted systemic absorption.

In recent years, drug retention at the corneal surface has been evaluated based on drug-loaded contact lenses [15,16]. For example, the delivery of pilocarpine by soaked contact lenses has been found to be ∼35 times more efficient than delivery by drops—a result matching clinical observations [17]. Drug loading and release studies demonstrated that introducing β-CD into hydrogels increased loading efficiency and achieved the sustained release of dexamethasone, as intraocular implants [18]. Although levels of up to 12.5% cyclodextrin are considered safe to use in eye drop formulations [19,20], the potential of using higher levels loaded into soft contact lenses (where there is no direct contact with the ocular surface) is unknown.

The use of cyclodextrin inclusion complexation and contact lenses are therefore both useful in enhancing ocular drug delivery. However, to the best of our knowledge, the potential efficacy of a combination of the two approaches in a composite system has not been explored. In this study, we aimed to test the hypothesis that enhanced topical delivery of econazole nitrate to the cornea could be achieved via a novel composite ocular drug delivery enhancement system comprising saturated drug/cyclodextrin-complex solutions loaded into hydrogel soft contact lenses, and that the doses delivered would be therapeutically relevant.

## 2. Materials and Methods

### 2.1. Materials

2′-hydroxypropyl-α-cyclodextrin and 2′-hydroxypropyl-β-cyclodextrin and Econazole nitrate were purchased from Sigma-Aldrich Company Ltd. (Poole, UK). HPLC-grade methanol and water, trifluoroacetic acid (TFA) and phosphate-buffered saline (PBS) tablets were purchased from Fisher Scientific UK Ltd. (Loughborough, UK). Acuvue TruEye^®^ daily disposable contact lenses (Johnson and Johnson, Ireland, UK) were provided by Cardiff University Eye Clinic (Cardiff, UK). Porcine eyeballs were supplied by a local abattoir having been enucleated immediately post-mortem.

### 2.2. Solubility Determinations

To determine the solubility of econazole nitrate, incremental amounts of drug were added to 1.5 mL of PBS in Eppendorf microcentrifuge tubes. These were then placed on a tube rotator overnight (room temperature, RT, or in an incubator at 37 °C), then visibly checked to ensure that econazole remained in excess, before being centrifuged for 15 min at 3500 RPM. Supernatants were sampled before being diluted with PBS and transferred to autosampler vials. Solubility was determined at room temperature and 37 °C.

To determine the effect of HP-α-CD and HP-β-CD on the solubility of econazole nitrate, different amounts of the cyclodextrin were added to Eppendorf tubes (up to 0.2 g per tube), and 1.5 mL of PBS was added. The tubes were shaken until the cyclodextrin had visibly dissolved, thereby providing a range of sub-saturated concentrations of HP-α-CD and HP-β-CD up to 0.133 g/mL). Next, econazole nitrate was added incrementally until no more would dissolve, the tubes were then placed in an ultrasonic bath for 30 min, and finally placed on a tube rotator overnight (RT and 37 °C), before processing as detailed above.

### 2.3. Corneal Delivery of Econazole In Vitro

The eyeballs, with cornea uppermost, were placed in a six Costar^®^ Well Cell Culture plate, and just enough PBS solution was added to cover half of the eyeballs [21]. The eyeballs were then placed in a water bath at 37 °C, providing a corneal surface temperature of 32 °C via heat dissipation. Next, 50 µL of econazole nitrate-saturated solution was dosed onto each eye using a pipette; dosing was repeated at 2 and 4 h, and the experiment stopped at 6 h. When comparing the efficacy of the saturated solution versus contact lenses, the solution was applied as a one-off dose, i.e., 150 μL, and left in the water bath for 6 h.

The contact lenses were loaded with drug–cyclodextrin using the ‘breathing in’ method [15]. Firstly, lenses were removed from their packaging, dried in an oven then placed in 10 mL of the relevant cyclodextrin–econazole-saturated solution overnight, set up in an incubator at 37 °C. The loaded lenses were then placed on the corneas of the upward-facing porcine eyeballs, which had been wetted beforehand with PBS. After 6 h, the lenses were removed and discarded, and each cornea was irrigated with 200 μL of PBS to remove excess drug on the surface. The corneas were then excised by blunt dissection using a scalpel, and subject to double-solvent extraction. The whole corneas were cut into small pieces (~1 mm^3^) and placed in a microcentrifuge tube containing 1 mL of methanol and left on a blood tube rotator overnight. Next, the cornea tissue was removed from the microcentrifuge tube and transferred to a new microcentrifuge tube containing a further aliquot of 1 mL of methanol, before being placed in an ultrasonic bath for 30 min and then rotated on a blood tube rotator for 1.5 h. Next, the tubes were centrifuged and the supernatant from the second extraction combined with that of the first, and the methanol was allowed to evaporate completely under vacuo at 60 °C. Extracted econazole was then reconstituted with 1 mL of methanol, centrifuged for 15 min, and the supernatant was filtered through a 0.45 μm syringe filter before being transferred to autosampler vials.

### 2.4. HPLC Analysis

Reverse-phase HPLC was used to quantitate the amounts of econazole in solubility and in vitro corneal determinations using an Agilent series 1100 system fitted with a Gemini-NX ODS 5 µ 110 Å 250 × 4.6 mm column (Phenomenex, Macclesfield, UK). The flow rate was 1 mL/min, the injection volume was 20 μL, and detection of econazole was carried out using UV at 230 nm. Analysis was performed at 20 °C, and under these conditions, econazole had a retention time of 3 min, with an overall run time of 8 min. Calibration curves of econazole nitrate were constructed over a concentration range of 0.977–500 μg/mL, giving a limit of quantitation of 0.2 μg/mL, and the mean linearity was R^2^ 0.9916.

### 2.5. Statistical Analysis

ANOVA (analysis of variance) and t-tests were carried out using InStat 3 statistical package for Macintosh (GraphPad Software Inc., San Diego, CA, USA) to compare the amount of drug released from eye drops, contact lenses and cyclodextrins to the corneas. A *p* < 0.05 was considered statistically significant. All graph plots and bar charts were constructed through the conversion of areas under the curves of each drug from HPLC analysis based on their calibration curves.

### 2.6. Molecular Dynamic Simulation

Molecular Operating Environment (MOE) 2015.10 [22] and Maestro (Schrödinger Release 2017-1) [23] were used as molecular modelling software. The initial HP-β-CD–econazole nitrate complex was built based upon that previously reported [24]. The complex was then energy-minimised using Amber 10 EHT force field in MOE2015.10. Econazole nitrate and HP-β-CD structures were downloaded from DrugBank [25,26] and energy-minimised. Molecular dynamics (MD) simulations were performed on Supermicro Intel^®^Xeon^®^ CPU ES-46200 @ 2.20 GHz × 12 running Ubuntu 14.04 using the Desmond package for MD simulation [27] with an OPLS-AA_2005 force field in ‘explicit’ solvent, employing the TIP3 water model. The initial coordinates for the MD simulation were taken from the previously prepared complexes. A cubic water box was used for the solvation of the system, ensuring a buffer distance of approximately 10 Å between each box side and the complex atoms. The system was minimised and pre-equilibrated using the default relaxation routine implemented in Desmond. A 100 ns MD simulation was performed, during which the equation of motion was integrated using a 2 fs time step in the NPT ensemble, with a constant temperature (300 K) and pressure (1 atm). All other parameters were set using the Desmond default values. Data were collected every 40 ps (energy) and every 160 ps (trajectory). HP-β-CD–ligand complex simulation was performed in triplicate, using a random seed every time as a starting point. Visualisation of the HP-β-CD–ligand complex and MD trajectory analysis was carried out using Maestro.

### 2.7. Mycological Evaluation

Walther et al. 2017 highlighted a number of fungal strains associated with cases of problematic keratitis [28]. Here, we carried out an in vitro evaluation of the potential efficacy of our econazole corneal delivery systems against a small panel of the strains highlighted, namely, *Fusarium semitectum, Fusarium solani, Scolecobasidium tshawytschae* and *Bipolaris hawaiiensis,* which were purchased from Public Health England, Porton Down, UK. Econazole nitrate was prepared at 1134 mg/L in sterile PBS and 0.1% Tween 20 (*w/v*) and stored at 30 °C for 12 h. Before use, the solution was vortex-mixed and checked for any undissolved powder. In columns 2–12 of a 96-well plate, 100 µL of PBS with 0.1% Tween 20 (*w/v*) was aliquoted. In columns 1 and 2, 100 µL of the econazole solution was added. After pipetting up and down to make sure the solution was mixed with the PBS–Tween 20, 100 µL from column 2 was placed into column 3. This was repeated across columns 3–10 to obtain doubling dilutions of econazole across the 96-well plate. Column 11 was used as a drug-free control.

All fungal strains in this study were grown on potato dextrose agar (PDA) at 30 °C. To obtain conidia, the two *Fusarium* strains were incubated for 48 h: *Fusarium bipolaris* for 72 h and *Fusarium scolecobasidium* for 120 h. Agar plates were then washed with 10 mL of PBS supplemented with 0.1% (*w/v*) Tween 20 and scraped with a sterile L-shaped spreader to collect conidia before filtration. The conidia suspensions were then centrifuged at 5500 rpm for 10 min before resuspending the pellet in 10 mL of (2×) RPMI 1640 media buffered with MOPS and 2% (*w/v*) glucose (pH 7). The suspension was then counted using a haemocytometer and adjusted in RPMI 1640 media to obtain 2 × 10^5^ cfu/mL. In columns 1–11 of the 96-well plate containing econazole, 100 µL of the suspension was added to give a final concentration of 1 × 10^5^ cfu/mL conidia. In column 12, containing only sterile PBS–Tween20, 100 µL of sterile (×2) RPMI 1620 was added and used as a sterile control. The 96-well plate was incubated at 30 °C for 48 h and the germination of conidia observed microscopically, with the minimum inhibitory concentration (MIC) being determined as the lowest concentration at which no germination occurred, as the mean of three experiments.

## 3. Results

### 3.1. Econazole Solubility

Econazole nitrate was found to have an aqueous solubility of 55.2 ± 4.4 μg/mL at 37 °C—literature values range from <1 mg/mL at 19 °C, and the online resource DrugBank [28] states 1.48 μg/mL (temperature not specified). In PBS, the solubility was significantly greater at 88.7 ± 6.1 μg/mL than water (*p* < 0.05). PBS is a mixture of electrolytes with considerably greater ionic strength than water, and it is known that increasing the ionic strength in an aqueous solution increases the solubility of a dissolved compound [29,30]. Further experiments continued with PBS solutions only.

Figure 2 and Table 1 show major dose-dependent increases in econazole solubility in the presence of sub-saturated levels of both HP-α-CD and HP-β-CD (*p* < 0.001) [31]. At first, increases in econazole solubility were high and proportionate at each increment of cyclodextrin added, but then tailed off in favour of excess cyclodextrin. Two trends were apparent: 1. econazole was significantly more soluble in the presence of both HP-α-CD and HP-β-CD at 37 °C compared to room temperature; and 2. econazole was significantly more soluble in the presence of HP-α-CD than HP-β-CD at both 37 °C and room temperature.

Table 1 also shows the M of econazole solubilised using HP-α-CD and HP-β-CD at RT and 37 °C, in relation to the M of the cyclodextrin present, along with the molar ratios thereof (M CD/M econazole). It is clear that despite the significant increases in econazole solubility per increment of cyclodextrin present, there was a large and increasing molar excess of cyclodextrin. Substantially more econazole was solubilised at 37 °C compared to room temperature, and substantially more econazole was solubilised using HP-α-CD than HP-β-CD, for example, ~50% more using HP-α-CD at 37 °C than HP-β-CD at 37 °C.

### 3.2. Corneal Delivery of Econazole in the Presence of Cyclodextrin and Contact Lenses

Figure 3 shows that an average of 21.66 μg econazole was delivered to the cornea from saturated solution. This was increased significantly (*p <* 0.001, factor of ×2) with the addition of HP-β-CD at 60.57 μg. There was a further increase when HP-α-CD was used, with a total of 87.65 μg (*p <* 0.001, factor of ×2.8). Figure 3 also shows that without cyclodextrin, an average 90.88 μg of econazole was delivered per cornea from the contact lens (*p <* 0.001). However, with HP-β-CD, this was significantly increased to an average of 160.17 μg per cornea (*p <* 0001, factor of ×1.76). There was a further increase when HP-α-CD was used, with a total of 200.8 μg (*p <* 0.001, factor of ×2.2). Compared to eye drops (21.66 μg), the enhancements were ×5 for HP-β-CD and ×6.34 for HP-α-CD. In summary, the highest levels of econazole were delivered to the cornea from a combination of contact lens and cyclodextrin.

### 3.3. Molecular Dynamic Simulations

In order to better understand the inclusion complexation process, a series of 100 ns molecular dynamics (MD) simulations were performed evaluating the behaviour of the different complexes in a water environment. Among the different inclusion models used as a starting point for the simulations, we examined two of them: the econazole–HP-β-CD complex in a 2:1 ratio with the two drugs inserted on the same side of the cyclodextrin (Figure 4A) and the econazole–HP-β-CD complex in a 2:1 ratio with the two drugs on the opposite side of the cyclodextrin (Figure 4B). Complex 4B was lost after a few ns of simulation with either both econazole molecules pushing each other out from the cyclodextrin cavity, causing the cyclodextrin to close on itself after a while (reducing the size of the central cavity), or one molecule of econazole remaining inside the cyclodextrin cavity, while the second molecule (Eco-2) hovered around outside and approached the complex from the same side, in a scenario similar to that seen for Figure 4A. Figure 4C is the final complex model for the complex shown in Figure 4B.

The distance between the two chlorine atoms from the mono-chloro phenyl ring of the two drugs (Cl1 and Cl2) and the distance between each chlorine atom and a cyclodextrin central oxygen atom were monitored during the MD simulation and are shown in Figure 5. The initial complex (Figure 4A) was stable for the first 5 ns of the simulation, as showed by the constant distance between the two chlorine atoms Cl1 and Cl2 and the distance between each chlorine atom and the cyclodextrin (state 1, Figure 5). Around 10 ns of simulation, Eco-1 was released into the water system while Eco-2 stayed in the complex, as highlighted by the increase in the Cl1-Cl2 and Cl1-CDX distance (state 2, Figure 5). Eco-1 then went back inside the complex (decrease and constant distances in the three graphs) and formed the 2:1 complex again for a few ns. Eco-1 went out and in again (changes in the distances), and then Eco-2 was released from the cyclodextrin (state 3, Figure 5), establishing an equilibrium in which a 2:1 ratio inclusion complex was altered to a 1:1 complex, with the two econazole molecules alternatively coming in and out from the cyclodextrin (state 4, Figure 5). This alternation is represented by the changes in the monitored distances; the increase in the Cl1-CDX and Cl1-Cl2 distance, and the contemporary low and constant Cl2-CDX distance, are indications that only Eco-1 had been released from CD. The increase in the Cl2-CDX and Cl1-Cl2 distance, and the contemporary low and constant Cl2-CDX distance, reflect that only Eco-2 was released from the cyclodextrin. Low and constant distances in all three graphs indicate 2:1 complex formation. Considering the results obtained for the complex A simulation, and the findings that the complex B was not stable during the simulations and its final trend was very similar to that of the complex A structure, the econazole–HP-β-CD inclusion complex formation in a 2:1 ratio with the drugs inserted on the same side of cyclodextrin was the most likely to form.

### 3.4. Antifungal Activity

The MIC data for the four fungal strains were found to be 141 μg/mL for both *Fusarium semitectum* and *Fusarium solani,* and 283 μg/mL for both *Scolecobasidium tshawytschae* and *Bipolaris hawaiiensis* (mean of three determinations). To determine potential efficacy, these figures were considered in relation to corneal volume. The volume of porcine corneas was determined as the difference between the native and dried (vacuum oven) weights of excised porcine corneas (*n* = 6), where the difference was water loss (g/cornea) and the equivalent of the amount of mL per cornea (mL/cornea)—this was found to be 0.19 mL.

Table 2 shows factors for the (econazole) delivered using the different drug delivery platforms at 6 h in relation to the (econazole) required to attain the MIC of the four fungal species. With 21.66 μg of econazole delivered to the cornea, the drops of the saturated solution achieved the least satisfactory result, producing sub-MIC levels at ×0.81 *Fusarium semitectum* and *Fusarium solani* and ×0.40 *Scolecobasidium tshawytschae* and *Bipolaris hawaiiensis*. All other treatments delivered (econazole) at > MIC for all four species.

Upon addition of HP-β-CD to the drops, factors of ×2.26 and ×1.13 were obtained for these species, respectively. However, when HP-α-CD was used, the levels were significantly increased to ×3.29 and ×1.63, respectively. The contact-lens-only delivery produced factors of ×3.39 and ×1.69. The composite HP-β-CD + contact lens produced factors of ×5.98 and ×2.98, respectively. However, the composite HP-α-CD + contact lens produced the highest factors of ×7.5 and ×3.73, respectively (Table 2, Figure 6). Figure 6 also highlights that the delivery of econazole was more efficacious against *Fusarium semitectum* and *Fusarium solani*, in terms of the higher factors achieved for these species compared to *Scolecobasidium tshawytschae* and *Bipolaris hawaiiensis.*

## 4. Discussion

Although natamycin tends to be the first-line treatment for fungal keratitis in numerous countries where the disease is prevalent, the broad-spectrum antifungal econazole nitrate has been proposed as a potential alternative. In a field trial, it was found that 2% econazole was equally as effective as 5% natamycin in treating fungal keratitis and, although the paper did not specify the vehicle used, these figures equate to 20 mg/mL and 50 mg/mL, respectively [7]. The observed efficacy of econazole nitrate may have been due to its lower molecular weight (381.7) than natamycin (665.7), which could have led to increased mass transport into the stroma, although other factors, such as relative potencies against causative fungi, need to be considered. Econazole nitrate has poor aqueous solubility, which can be increased by co-solvency using organic solvents, although this is not conducive to ocular drug delivery [32]. In a fluid medium, the antimycotic effect of the econazole β-CD complex against a strain of *Candida albicans* was superior to the effect of a physical mixture of the two compounds, and a small antimicrobial effect was noted [33].

Here, we focussed on the combined techniques of cyclodextrin complexation and loaded contact lenses to achieve enhanced ocular drug delivery. Combination approaches are not new in the field of ocular drug delivery, but the use of cyclodextrins to increase drug solubility prior to loading contact lenses is novel. Importantly, the contact lenses used are composed of silicone hydrogel, a material which, importantly, allows almost 100% of available oxygen to pass through to the corneal surface (Acuvue) [34].

### 4.1. Solubility Enhancement by Cyclodextrins

Cyclodextrins are generally regarded as safe and are well-recognised for their ability to increase the aqueous solubility of a drug and are included in ocular drug formulations not only to increase aqueous drug solubility, but also to enhance drug absorption into the eye, increase aqueous stability and reduce local irritation. Examples of eye drop products that contain drug–cyclodextrin complexes on the market are indomethacin (Indocid^®^), employing 2% HP-β-CD, chloramphenicol (Clorocil^®^), employing RM-β-CD, and diclofenac sodium (Voltaren ophtha^®^), employing HP-γ-CD as an excipient in their formulations [10]. Hydroxypropyl derivatives of cyclodextrin have greater water solubility, and HP-β-CD is the most commonly employed cyclodextrin in eye drop formulations [11]. Both HP-α-CD and HP-β-CD are themselves both highly water-soluble; Chemicalbook [31] recorded the aqueous solubility of HP-α-CD and HP-β-CD at 600 and 450 mg/mL, respectively. Thus, even the highest levels used here (0.132 g/mL) represent a small percentage of their solubilities (~0.02%). In terms of cyclodextrin tolerability, HP-β-CD at concentrations up to 12.5% w/v (125 mg/mL) was found to be well-tolerated on the corneal surface of rabbit eyes [20]. On the other hand, previous work using excised bovine eyes suggests α-CD might interfere with membrane structure, especially with the lipoidal epithelial cell layers causing barrier destabilisation [35], although no similar effect has been reported for HP-α-CD.

The pH of the eye surface ranges from 7 to 7.4, and the pH of an instilled solution could affect ophthalmic drug ionisation and subsequent absorption [11]; therefore, we used PBS as econazole has a Pka of 6.77 (Drug bank). The eye temperature in acute keratitis is slightly raised from ~35 °C and ranges from 36.05 °C to 38.60 °C [36]. Hence, in this study, the solubility of econazole nitrate in the presence of cyclodextrin was tested at 37 °C versus room temperature for comparison.

The remarkable ability of cyclodextrin to increase the solubility of poorly soluble drugs is clear in the current work, as shown in Figure 2. The higher the amount of cyclodextrin added into the PBS solution, the higher the concentration of econazole found in the PBS solution. The incremental increases in solubility tended to tail off as the amount of cyclodextrin approached 13.2%. This could be due to an excess complexation of drug with cyclodextrins. At first, a clear cyclodextrin solution was produced, which became cloudy when an excess econazole nitrate was added to the mixture. It was likely that at such high concentrations, the precipitate consisted of both excess crystalline econazole nitrate [37] and cyclodextrin. Precipitation of cyclodextrin in this manner would mean reduced solubilisation potential for econazole, possibly contributing to the loss of linearity as seen in Figure 2.

### 4.2. The Relationship between Solubility Enhancement of Econazole and Amount of Added HP-α-CD and HP-β-CD

Closer examination of the relationship between solubility enhancement of econazole and amount of cyclodextrin added revealed a considerable molar excess of HP-α-CD and HP-β-CD relative to econazole. Let us consider the results for 13.2% *w/v* HP-α-CD (or 1.19 × 10^−1^ M) in PBS. This gave an enhanced econazole solubility of 2293 µg/mL or 6.0 × 10^−3^ M. Thus, the molar ratio was 19.23 HP-α-CD:econazole; i.e., for every molecule of econazole in solution, there were 20 HP-α-CD molecules. At this stage, i.e., at saturation, the PBS cannot accommodate any more molecules of econazole, despite the availability of excess HP-α-CD. The answer might lie in the fact that a molecule of econazole must diffuse across a relatively vast expanse of PBS, in which it has poor solubility, for it to be able to interact with HP-α-CD. Increasing the level of HP-α-CD increases the probability of interacting with econazole molecules, thereby increasing solubility but never, due to its hydrophobicity, to 1:1. Such 1:1 molar ratios have been reported using kneading, co-precipitation and freeze-drying complexation techniques [14].

A 1:2 ratio of CD and econazole was reported by [24], and this was also confirmed by our molecular modelling studies (Figure 4). Various molecular ratios of econazole nitrate to HP-β-CD were investigated through running a series of 100 ns molecular dynamics (MD) simulations. Results show an optimum of a 2:1 complex (econazole:HP-β-CD), with two molecules of econazole nitrate entering the same side of the cavity of HP-β-CD. However, in a saturated solution system, the situation is far more complicated. A non-linear plot with concave downward curvature may be evidence of non-ideality effects (non-constancy of activity coefficient) or of self-association by the ligand. Such an excess of cyclodextrin in relation to solubilisate was also previously reported for acetazolamide [10] and econazole [9]. Phase diagrams do not always accurately reflect the dynamic nature of guest–host association–dissociation. Loftsson et al. (2005) [10] reported that the solubilising effects of cyclodextrins are due to the formation of multiple inclusion complexes and non-inclusion interactions, or simpler non-interaction admixtures—attempts to model this were unsuccessful in the current work. This is further complicated in that drug–cyclodextrin complexes can self-associate to form water-soluble aggregates, which then can further solubilise the drug through non-inclusion complex interactions [38].

Regardless, complex *dissociation* is required at the cornea interface to allow the drug to penetrate into the cornea, but there is no evidence in the literature to suggest that cyclodextrins do this. In an aqueous solution, drug–cyclodextrin complexes are continually associating and dissociating with lifetimes in the range of milliseconds or less, and although stronger association would result in slower release kinetics, the rates are still essentially instantaneous [39]. The dynamic association–dissociation of econazole in cyclodextrin was evident during our molecular dynamic simulations (Figure 4 and Figure 5). Under such circumstances, econazole released on the ocular surface from topically applied HP-α-CD and HP-β-CD would become bioavailable and establish elevated concentration gradients and diffusion into the corneal epithelium in accordance with Fick’s law. This assumes that HP-α-CD and HP-β-CD are not affecting the barrier function of the cornea.

### 4.3. Effect of HP-β-CD on the Corneal Delivery of Econazole

Porcine eyes were chosen as the animal model in this study as porcine eyes have been found to share many similarities with human eyes in terms of biochemical properties [40]. HP-β-CD is already used in a number of ocular drug delivery products and has been shown to be non-irritating in rabbit eyes after application of eye drops containing only pure aqueous 45% *w/v* HP-β-CD solution (Loftsson and Järvinen 199) [11]. However, the maximum concentration of HP-β-CD that can be tolerated by the cornea is reported to be 12.5% [41]. Therefore, a level of 10% *w/v* HP-α-CD and HP-β-CD was chosen for this work.

The result from the experiment performed on porcine corneas did not reflect that from the solubility data. Here, we saw a slightly greater (although not statistically significant) concentration of econazole delivered to the cornea from HP-β-CD than that from HP-α-CD. Clearly, the formation of inclusion complexes alone does not explain the corneal delivery data. Dissociation of the complex at the cornea surface is essential in allowing the drug to penetrate the epithelium and diffuse into the stroma. Having a more restricted cavity volume, HP-α-CD probably formed a tighter inclusion complex with econazole relative to HP-α-CD, resulting in inhibited release. A further factor relates to the effect of cyclodextrin at the corneal surface, involving the sequestration of cholesterol from the epithelium, causing disruption of its integrity, and rendering it more permeable to the penetration of econazole. Molecular dynamics simulations were previously used to demonstrate that cyclodextrins have a strong affinity to bind to a membrane surface, and by doing so, destabilise the local packing of cholesterol molecules, making their extraction favourable and rendering the membrane leaky [42].

In the current work, the significantly greater solubilising capability of HP-α-CD may have been counteracted by its reduced propensity to sequester cholesterol from the corneal epithelium, resulting in a delivery of econazole that was comparable to the use of HP-β-CD. Abd El-Gawad et al., 2017 [14] showed that all eye drops containing co-precipitate complexes exhibited a higher release rate than that of other complexes and followed the diffusion-controlled mechanism. An in vivo study proved that eye drops containing econazole–CD complexes showed higher ocular bioavailability than EC alone, which was indicated by the higher AUC, C_max_ and relative bioavailability values.

Our modelling work clearly shows the dynamic association–dissociation of econazole in cyclodextrin, with econazole being released into the solution (Figure 5). If this occurred within the bulk fluid phase away from the corneal surface, the release was followed by reformation of the 2:1 inclusion complex shown. However, if release occurred adjacent to, or in contact with, the ocular surface, the molecules of econazole would become momentarily bioavailable and, through the increased concentration gradient, provided increased diffusion into the cornea in accordance with Fick’s law. This is summarised in Figure 7.

### 4.4. Mechanism for the Enhanced Corneal Delivery Econazole Using the Composite Contact Lens–HP-α-CD System

The use of drug-loaded contact lenses as vectors for ocular drug delivery has made this a popular approach to produce new medicines for a range of eye diseases [43], as exemplified by the recent launch of ACUVUE^®^ Theravision™, which is a daily disposable contact lens (etafilcon A) loaded with the antihistamine ketotifen. Both silicone hydrogel and pHEMA hydrogel contact lenses have been used to good effect for vitamin E, timolol, dexamethasone and dexamethasone 21-acetate [16,44]. In the current work, econazole-loaded hydrogel contact lenses were also shown to successfully deliver more econazole compared to eye drops (Figure 3). However, our approach involved a further step were we used solutions of econazole–10% *w/v* cyclodextrin to impregnate commercially available soft hydrogel contact lenses, thereby producing a composite ocular drug delivery system; composite hydrogel systems are a growing and exciting drug delivery research area [45,46]. The increase in delivery of econazole to the corneas using econazole–HP-β-CD inclusion complexes, as mentioned above and as seen in Figure 4, was significantly increased again when they were used to load hydrogel contact lenses. For HP-α-CD, ~130 μg of econazole was delivered to the cornea, whereas for HP-β-CD, the amount delivered was significantly higher at ~160 μg (*p <* 0.05).

However, this concentration of drug extracted from the cornea did not reflect the solubility data, which showed a higher concentration. Cyclodextrin is believed to remain in the aqueous solution while helping to deliver hydrophobic drugs to the surface of the biological membrane. It is believed that only free drug that dissociates from the drug–cyclodextrin complex can enter the biological membranes [10,41]. As the contact time between the econazole–cyclodextrin complex and the eye after the instillation of eye drops is short due to the washing out of eye drops by tears and the drainage system, the cyclodextrins may not have enough time to release all drugs from the cavity to the eye surface. Hence, the idea of delivering drug from contact lens and cyclodextrin would be beneficial.

Under normal operating conditions, contact lenses do not in fact make contact with the eye; rather, they should float upon a thin film of tear fluid. The tear film provides an additional diffusional barrier for lipophilic drugs applied topically to the cornea. The significance of the aqueous tear film between the contact lens and the ocular surface and the subsequent drug uptake by the cornea has been modelled previously [17]. In confirming that drug delivery from a contact lens is more efficient than drug delivery by drops, it was also shown that corneal penetration was much reduced by the presence of the tear film. Such effects would be affected by drug lipophilicity, and the partitioning of lipophilic econazole into the aqueous tear film would be inhibited. However, the presence of the HP-β-CD within the formulation would act as a transporter to facilitate the mass transfer of drug molecules between the contact lens and the ocular surface.

As HP-α-CD is smaller than HP-β-CD, i.e., it has a lower molecular volume, more of it can be accommodated within the matrix of the hydrogel. Similarly, the smaller size of HP-α-CD means it and its drug load should be more easily liberated from the lens matrix, due to reduced steric hindrance. These effects combine to provide the highest corneal penetration in the case of lenses loaded with econazole and HP-α-CD.

### 4.5. Clinical Considerations

In the study by Prajna et al. [6], the majority of cases of fungal keratitis were found to be caused by *Fusarium* spp., against which econazole has a reported MIC value of 8 mg/L [47,48]. In the current work, with both *Fusarium semitectum* and *Fusarium solani,* econazole gave considerably higher MICs of 141 μg/mL (141 mg/L)—approximately ×17 higher. Other organisms successfully treated by econazole include *Aspergillus flavus* (MIC 4 mg/*L* [49]), *A. fumigatus* (MIC 8 mg/L [48]) and non-sporulating hyaline moulds (MIC 4 mg/L [47]), the most econazole-resistant of which was *A. fumigatus.* In fact, the highest econazole MIC reported in the literature for any fungal keratitis isolate is 8 mg/L. In the current work, the other two fungal strains tested were *Scolecobasidium tshawytschae* and *Bipolaris hawaiiensis*, against which econazole again gave comparatively high MICs of 283 μg/mL for both. The reasons for the higher values in the current work are unclear.

Based on a typical corneal volume of 0.19 mL [50], Table 2 shows that the basic drops are unlikely to be able to deliver a therapeutic level of econazole to corneas infected with fungal infiltrates. The composite system of cyclodextrin plus contact lens produced the highest efficacy factors, demonstrating the enhanced efficacy of the composite approach to corneal drug delivery. However, although few clinical trials have been reported, econazole appears to be well-tolerated during extended ocular use of 1 or 2% preparations in humans [6,50] and has historically been used in the UK as an alternative to natamycin for filamentary fungal infections [51,52,53]. Furthermore, no significant cytotoxicity was observed in primary human corneal epithelial cell cultures after 7 or 14 days of exposure to up to 2500 mg/L econazole in vitro [54]—more than 4 times the estimated concentration delivered by the present formulation. It should also be borne in mind that cyclodextrins are reported to mitigate toxic effects of the drugs loaded into them, e.g., diclofenac [55], by essentially presenting a barrier between the drug and the corneal surface. Furthermore, doses of HP-β-CD at a concentration of 12.5% were found to be well-tolerated and non-toxic to the corneal epithelium when evaluated using SLB and SEM [56].

In this study, single doses of loaded contact lenses and eye drops were applied on porcine corneas over the timescale of 6 h; this generally reflects overnight application, which, although not normally recommended, may be defensible for the administration of critical medication for such a sight-threatening condition.

## 5. Conclusions

At saturation, econazole formed complexes with HP-α-CD to a greater extent than HP-β-CD, but both provided greatly increased drug solubility in excess of a 2:1 ratio, indicating complexity in the solubilisation mechanism. When incorporated into the contact lens matrix, high-concentration gradients could be established overlying the cornea with higher drug bioavailability due to the transient release of drug from the cyclodextrin, and dose retention on the ocular surface. The composite system involving contact lenses and HP-α-CD delivered higher doses of econazole into the cornea compared to that of HP-β-CD. Importantly, the drug levels delivered by both composite systems were found to be therapeutically relevant against four major causative organisms involved in fungal keratitis. Overall, the data suggest that superior clinical improvements could be obtained using this composite system in cases of this intractable and debilitating disease.

## Figures and Tables

**Figure 1 pharmaceutics-14-01631-f001:**
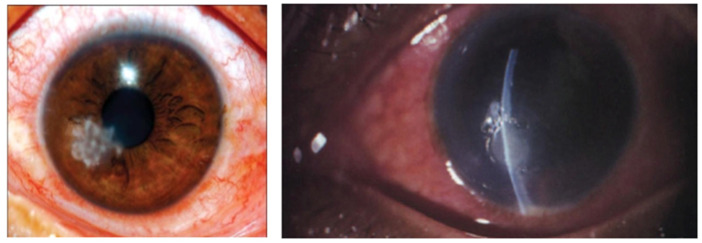
Images showing stromal fungal infiltrates causing fungal keratitis.

**Figure 2 pharmaceutics-14-01631-f002:**
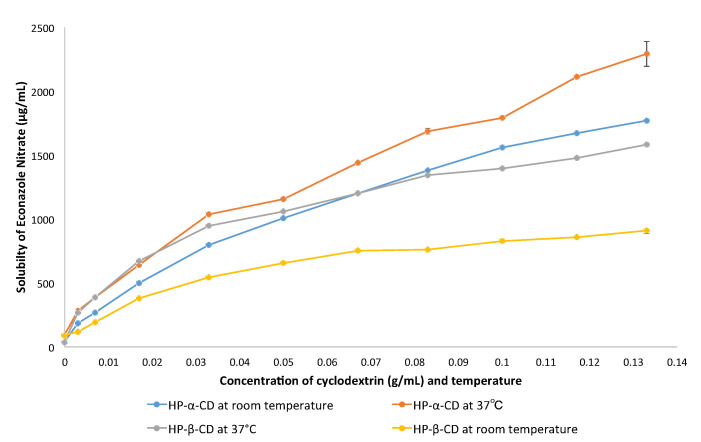
Solubility of econazole nitrate (g/mL) in the presence of sub-saturated levels of HP-α-CD and HP-β-CD in PBS at room temperature and 37 °C.

**Figure 3 pharmaceutics-14-01631-f003:**
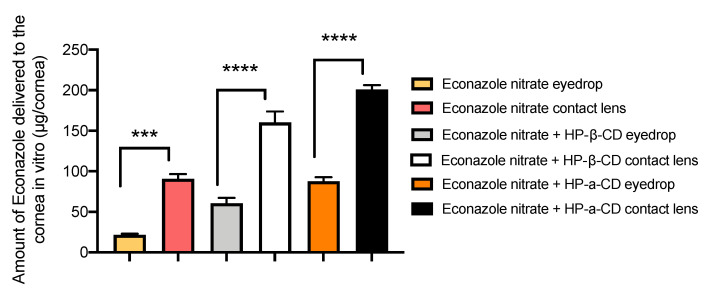
Mass of econazole delivered to porcine corneas (μg/cornea) after 6 h, *n* = 3 ± SD. Statistically significant differences indicated by *, where *** *p* < 0.001 and **** *p* < 0.0001.

**Figure 4 pharmaceutics-14-01631-f004:**
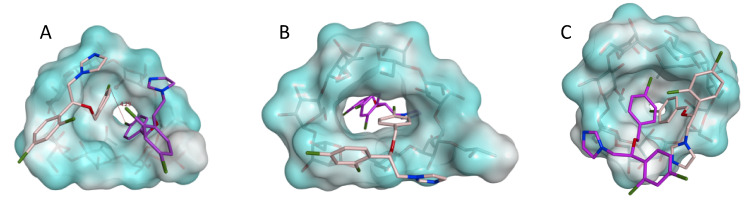
Two proposed inclusion complex models: econazole–HP-β-CD complex in a 2:1 ratio with the two drugs entering the same side of the cyclodextrin (**A**), and econazole–HP-β-CD complex in a 2:1 ratio with the two drugs on the opposite side of the cyclodextrin (**B**). Econazole carbon atoms are in pink (Eco-1) and purple (Eco-2). HP-β-CD is represented as a molecular surface. (**C**) is the final complex model for complex 4B. Eco-1 (pink) remained inside the cyclodextrin while Eco-2 (purple) approached the complex from the same side. Water molecules from the simulation environments have been removed for the clarity of the image.

**Figure 5 pharmaceutics-14-01631-f005:**
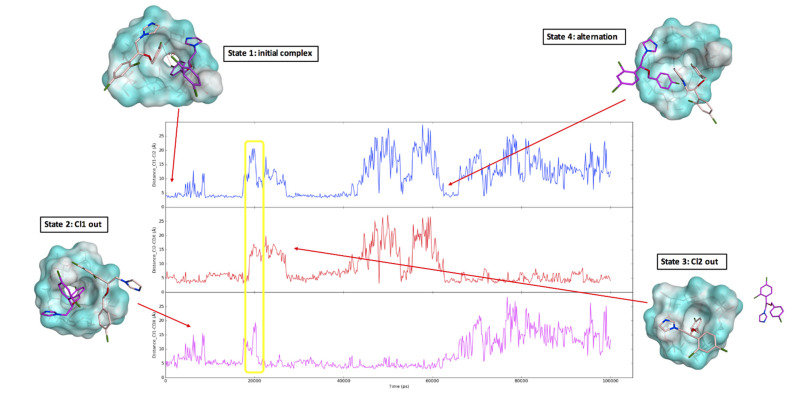
Molecular dynamic simulation results. The three graphs show the distance between the two chlorine atoms from the mono-chloro phenyl ring of the two drugs (Cl1 and Cl2, blue line) and the distance between each chlorine atom and a CD central oxygen atom (Cl1-CDX, purple line, Cl2-CDX, red line). Four different states were observed during the simulations: state 1, a 2:1 complex in which the three monitored distances were constant; state 2, in which Eco-1 (Cl1) was released from the cyclodextrin (with increases in the Cl1-CDX (purple line) and Cl1-Cl2 (blue line) distances); state 3, in which Eco-2 (Cl2) was released from the cyclodextrin (with increases in the Cl2-CDX (red line) and Cl1-Cl2 (blue line) distances); state 4: alternation between the two econazole molecules occupying the cyclodextrin cavity (in this specific moment of the simulation, Eco-2 was approaching the cyclodextrin (decrease in Cl2-CDX distance—red line), while Eco-1 was slowly coming out (increase in the Cl1-CDx distance—purple line)). The yellow circle highlights a brief simulation instance in which the two econazole molecules were both outside of the cyclodextrin cavity. Distance (Y axis) was measured in Å, time (X axis) in ps. Econazole carbon atoms are in pink (Eco-1) and purple (Eco-2). HP-β-CD is represented as a molecular surface. Water molecules from the simulation environments have been removed for the clarity of the image.

**Figure 6 pharmaceutics-14-01631-f006:**
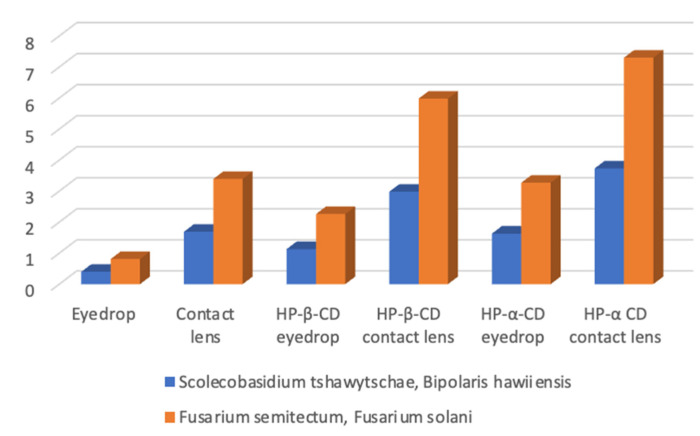
Efficacy of different platforms for the delivery of econazole to the cornea in relation to the MIC of econazole against four fungal species associated with fungal keratitis, where 1 = MIC (the MICs of *Scolecobasidium tshawytschae* and *Bipolaris hawaiiensis* were the same, as were those of *Fusarium semitectum* and *Fusarium solani*). Eye drops generally delivered sub-MIC levels, whereas HP-α-CD delivered 3.5–7.1 ×MIC.

**Figure 7 pharmaceutics-14-01631-f007:**
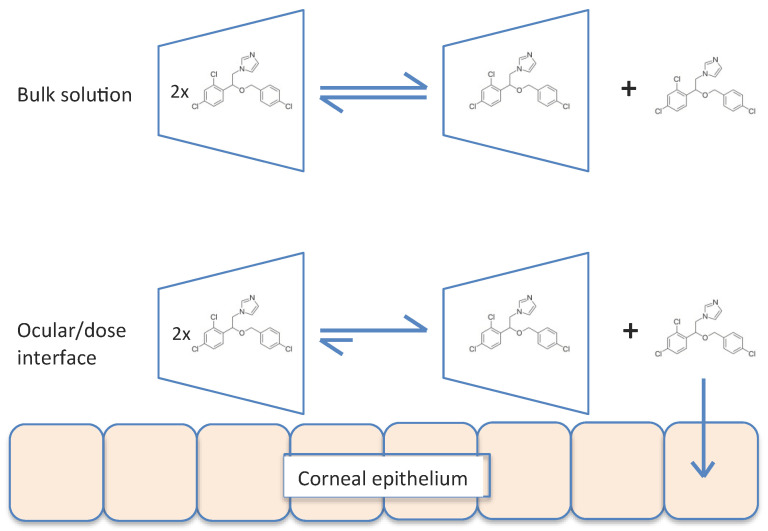
Equilibrium of the association–dissociation of econazole–cyclodextrin inclusion complexes in bulk liquid phase of the applied dose (**upper**), and uptake of transiently free econazole when adjacent to or in contact with the corneal interface (**lower**).

**Table 1 pharmaceutics-14-01631-t001:** Solubility of econazole with increasing amounts of HP-α-CD or HP-β-CD in PBS at room temperature and 37 °C, and molar ratios of cyclodextrin–econazole in saturated PBS solution. MW HP-α-CD = 1180, HP-β-CD = 1375, econazole nitrate = 381.683 (to convert g/mL to M, multiply concentration by 10^3^/MW).

g/mL CD	M HP-α-CD	Room Temperature			37 °C			M HP-β-CD	Room Temperature			37 °C		
		μg/mL Econazole	M Econazole	M CD/M Econazole	μg/mL Econazole	M Econazole	M CD/M Econazole		μg/mL Econazole	M Econazole	M CD/M Econazole	μg/mL Econazole	M Econazole	M CD/M Econazole
**0**	**0**	38.03	2.33 × 10^−4^	-	91.35	4.11 × 10^−4^	-	**0**	88.73	2.33 × 10^−4^	-	156.91	4.11 × 10^−4^	-
**0.0033**	**2.80 × 10^−3^**	185.29	4.85 × 10^−4^	5.77	280.18	7.34 × 10^−4^	3.81	**2.40 × 10^−3^**	119.77	3.14 × 10^−4^	7.64	268.09	7.02 × 10^−4^	3.42
**0.0066**	**5.59 × 10^−3^**	268.14	7.49 × 10^−4^	7.46	384.02	1.00 × 10^−3^	5.59	**4.80 × 10^−3^**	192.33	5.04 × 10^−4^	9.52	386.43	1.01 × 10^−3^	4.75
**0.0165**	**8.47 × 10^−3^**	497.65	1.30 × 10^−3^	6.51	638.27	1.67 × 10^−3^	5.07	**1.82 × 10^−2^**	378.73	9.92 × 10^−4^	18.37	668.76	1.75 × 10^−3^	10.4
**0.033**	**2.79 × 10^−2^**	800.65	2.10 × 10^−3^	13.29	1038.51	2.72 × 10^−3^	10.36	**2.40 × 10^−2^**	547.47	1.43 × 10^−3^	16.78	950.05	2.49 × 10^−3^	9.64
**0.05**	**4.24 × 10^−2^**	1010.17	2.65 × 10^−3^	16.00	1155.01	3.30 × 10^−3^	12.85	**3.64 × 10^−2^**	657.45	1.72 × 10^−3^	21.16	1062.3	2.78 × 10^−3^	13.09
**0.066**	**5.59 × 10^−2^**	1199.41	3.14 × 10^−3^	17.80	1438.15	3.77 × 10^−3^	14.83	**4.80 × 10^−2^**	755.42	2.03 × 10^−3^	23.65	1200.61	3.14 × 10^−3^	15.29
**0.0825**	**6.99 × 10^−2^**	1383.41	3.62 × 10^−3^	19.31	1689.42	4.43 × 10^−3^	15.78	**6.00 × 10^−2^**	762	2.00 × 10^−3^	30.00	1343.19	3.52 × 10^−3^	17.05
**0.1**	**8.47 × 10^−2^**	1561.05	4.09 × 10^−3^	20.71	1794.04	4.70 × 10^−3^	18.02	**7.27 × 10^−2^**	831.69	2.18 × 10^−3^	33.35	1398.24	3.66 × 10^−3^	19.86
**0.1155**	**9.79 × 10^−2^**	1673.3	4.38 × 10^−3^	22.35	2116.84	5.54 × 10^−3^	17.67	**8.40 × 10^−2^**	861.6	2.26 × 10^−3^	33.17	1479.49	3.88 × 10^−3^	21.65
**0.132**	**1.19 × 10^−1^**	1769.13	4.63 × 10^−3^	25.70	2293.58	6.00 × 10^−3^	19.83	**9.60 × 10^−2^**	908.56	2.37 × 10^−3^	40.51	1584.94	4.15 × 10^−3^	23.13

**Table 2 pharmaceutics-14-01631-t002:** Minimum inhibitory concentrations (MICs) of econazole against four strains associated with fungal keratitis, the concentration required in the cornea to attain MIC, the concentration of econazole localised in the cornea after 6 h (*n* = 3 ± SD) and factor (based on a corneal volume of 0.19 mL).

Fungal Strain	*Fusarium* *semitectum*	*Fusarium* *solani*	*Scolecobasidium* *tshawytschae*	*Bipolaris* *hawaiiensis*
Mean MIC [econazole], μg/mL	141	141	283	283
[Econazole] required in cornea (MIC×0.19), μg	26.79	26.79	53.77	53.77
[Econazole] in cornea, @ 6h; econazole only eyedrop	21.66 ± 2.36	21.66 ± 2.36	21.66 ± 2.36	21.66 ± 2.36
Factor, [econazole] delivered/MIC×0.19	×0.81	×0.81	×0.40	×0.40
[Econazole] in cornea, @ 6h; econazole contact lens	90.88 ± 9.75	90.88 ± 9.75	90.88 ± 9.75	90.88 ± 9.75
Factor, [econazole] delivered/MIC×0.19	×3.39	×3.39	×1.69	×1.69
[Econazole] in cornea, @ 6h; econazole + HP-β-CD eyedrop	60.57 ± 11.47	60.57 ± 11.47	60.57 ± 11.47	60.57 ± 11.47
Factor, [econazole] delivered/MIC×0.19	×2.26	×2.26	×1.13	×1.13
[Econazole] in cornea, @ 6h; econazole + HP-β-CD contact lens	160.17 ± 23.63	160.17 ± 23.63	160.17 ± 23.63	160.17 ± 23.63
Factor, [econazole] delivered/MIC×0.19	×5.98	×5.98	×2.98	×2.98
[Econazole] in cornea, @ 6h; econazole + HP-α-CD eyedrop	87.65 ± 8.74	87.65 ± 8.74	87.65 ± 8.74	87.65 ± 8.74
Factor, [econazole] delivered/MIC×0.19	×3.27	×3.27	×1.63	×1.63
[Econazole] in cornea, @ 6h; econazole + HP-α-CD contact lens	200.8 ± 9.6	200.8 ± 9.6	200.8 ± 9.6	200.8 ± 9.6
Factor, [econazole] delivered/MIC×0.19	×7.5	×7.5	×3.73	×3.73

## Data Availability

Information on the data underpinning the results presented here, including how to access them, can be found in the Cardiff University data catalogue at http://doi.org/10.17035/d.2022.0215856644.

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
