# Peer review of "A Composite System Based upon Hydroxypropyl Cyclodextrins and Soft Hydrogel Contact Lenses for the Delivery of Therapeutic Doses of Econazole to the Cornea, In Vitro"

_pharmaceutics, 2022, doi:10.3390/pharmaceutics14081631_

Round 1

Reviewer 1 Report

The manuscript written by Wong et al. is about engineering materials for ocular delivery, specifically, for the treatment of fungal keratitis. The authors developed a strategy based on the preparation of cyclodextrins and soft hydrogel contact lenses for the delivery of econazole. The authors found remarkable effects when combining cyclodextrines and soft hydrogels. The manuscript is interesting and well organized. The results are convincing and experimental data contain appropriate control experiments with a detailed experimental section. I think this manuscript may be publishable in Pharmaceutics, but a couple of issues should be explained by the authors.

 -          Figure 6. The graphic should include error bars in each measurement. The legend should include the SD (SD = n?) 

-     The authors have reported that a soft hydrogel was prepared. It would be convenient to further characterize this system including rheological properties like the evolution of the storage (G’) and the loss (G’’) moduli as a function of temperature.  

Author Response

- Figure 6. The graphic should include error bars in each measurement. The legend should include the SD (SD = n?) 

Error bars are not appropriate in this case. We have amended the legend, which is now hopefully clearer.

-     The authors have reported that a soft hydrogel was prepared. It would be convenient to further characterize this system including rheological properties like the evolution of the storage (G’) and the loss (G’’) moduli as a function of temperature.  

 The contact lenses were not synthesised - we used were a proprietary brand: Acuvue TruEye® daily disposable contact lenses (Johnson and Johnson, Ireland). We dont feel that the suggested physical examination would add to the main message in the manuscript.

Reviewer 2 Report

The manuscript mainly reported an ocular drug delivery system comprised of econazole/cyclodextrin complex saturated solutions loaded into hydrogel soft contact lenses. Econazole formed complexes with HP-α-CD and HP-β-CD, leading to greatly increased drug solubility with complex solubilization mechanisms. When incorporated into the contact lens matrix, greater corneal drug delivery could be achieved.

The composite drug delivery system enhanced topical delivery of econazole nitrate to the cornea, and the doses delivered would be therapeutically relevant against four major causative organisms involved of fungal keratitis. However, some data were not reasonably explained. There was a mismatch between some data and interpretation. Therefore, I think this paper can only be reconsidered after fully addressing several major issues as noted below:

1.    The description of Figure 3 in the page 10, line 372 to line 376, was actually from the content of Table 2 and Figure 6.

2.    In the page 7, line 266 to line 267, what were the data and basis supporting the final conclusion that “HP-α-CD at RT stands out as providing the greatest Solubility enhancement”?

3.    The maximum data of 0.2µg/mL in the abscissa of Figure 2 in the page 6 didn’t correspond to “up to 13.2% w/v, (actual experiment involved 0.2g in 1.5ml)” in the page 4, line 135. Therefore, part of the corresponding description of Figure 2 was inaccurate. And the unit “µg/mL” of the ordinate in Figure 2 didn’t match the unit “g/mL” in the caption.

4.    The content of Section 3.1 was confusing. The description of Table 2 and Table 3 in the page 7 was incorrect because the description didn’t match Table 2 and the manuscript didn’t have Table 3.

5.    The last section “Conclusion” could be numbered. And it would be more complete to distinguish HP-α-CD and HP-β-CD in the conclusion; for example, some conclusions contained only HP-β-CD data.

6.    Those wavy green lines in Table 1 in the page 7 and Table 2 in the page 10 made the table seem less standard.

7.    The manuscript should provide the full name when an abbreviation first appeared, such as “MIC” in the page 1, line 24.

8.    The formats of the figures in the manuscript were not uniform, such as the position of the caption in Figure 4 in the page 8.

9.    The authors could consider adding the following articles into references which would again increase the interest to general composite hydrogel readers: Chemical Society Reviews, 2021, 50, 8319-8343; Advanced Functional Materials‚ 2022‚ 32, 2108749; Biomaterials Science ‚ 2020‚ 8, 4940-4950.

Author Response

The description of Figure 3 in the page 10, line 372 to line 376, was actually from the content of Table 2 and Figure 6.

Thank you for pointing out this error - it has now been corrected.

In the page 7, line 266 to line 267, what were the data and basis supporting the final conclusion that “HP-α-CD at RT stands out as providing the greatest Solubility enhancement”?

line 259-267 - we apologise, this is an error, and is a remnant from an earlier draft.

The maximum data of 0.2µg/mL in the abscissa of Figure 2 in the page 6 didn’t correspond to “up to 13.2% w/v, (actual experiment involved 0.2g in 1.5ml)” in the page 4, line 135. Therefore, part of the corresponding description of Figure 2 was inaccurate. And the unit “µg/mL” of the ordinate in Figure 2 didn’t match the unit “g/mL” in the caption.

Again we apologise for this oversight. These errors have been corrected. The overall message remains unchanged.

The content of Section 3.1 was confusing. The description of Table 2 and Table 3 in the page 7 was incorrect because the description didn’t match Table 2 and the manuscript didn’t have Table 3.

line 259-267 have been deleted (this was relevant to a prior draft); new description added relating to Table 1. 

The last section “Conclusion” could be numbered. And it would be more complete to distinguish HP-α-CD and HP-β-CD in the conclusion; for example, some conclusions contained only HP-β-CD data.

The relevant changes have been made

Those wavy green lines in Table 1 in the page 7 and Table 2 in the page 10 made the table seem less standard.

I believe this to be a microsoft word spell checking feature, which will not appear in the final article

The manuscript should provide the full name when an abbreviation first appeared, such as “MIC” in the page 1, line 24.

Done

The formats of the figures in the manuscript were not uniform, such as the position of the caption in Figure 4 in the page 8.

Done

The authors could consider adding the following articles into references which would again increase the interest to general composite hydrogel readers: Chemical Society Reviews, 2021, 50, 8319-8343; Advanced Functional Materials‚ 2022‚ 32, 2108749; Biomaterials Science ‚ 2020‚ 8, 4940-4950.

The second of these suggestions has been added.

Reviewer 3 Report

Abstract:

“Hydroxypropyl-α-cyclodextrin (HP-α-CD) and hydroxypropyl-β-cyclodextrin (HP-β-CD) were added incrementally to deionized water or PBS prior to the addition of excess econazole nitrate, and the saturated solubility determined following centrifugation and analysis of the supernatant by reverse phase HPLC”. I think no need for this in the abstract.

“The MIC of econazole was determined against four fungal species associated with keratitis, from which [econazole] required to attain MIC/[econazole] were determined”. What is the solution-only drops ? it is better to use eyedrops.

Introduction:

Overall, there is no proper flow and good story in the introduction

Some examples

The introduction part is introducing the disease without any prevalence at least in authors country and without financial burden, and it did not introduce the drug very well. The introduction did not answer the principal question– can we use econazole for ocular delivery? If it is safe, why there is no single approved formulation for ocular applications.

“Natacyn® 5% (natamycin) suspension eye drops and is the first-line topical ophthalmic antifungal preparation in many countries”. 

“The drug has poor penetration into the cornea and the extent of infection can vary widely, therefore duration of treatment can take from a few weeks to months”. It seems unrelated to be included in one sentence.

“Increasing the dose of drug in the formulation can result in improved bioavailability and therapeutic outcomes in accordance with Fick’s first law of diffusion, and this should be achievable by changing the drug delivery platform”. This general nonscientific talk and if so needs to be substantiated by references.

“The potential of cyclodextrins to increase the solubility of econazole is not a new concept [9,13,14], but although improved therapeutic outcomes may result from increased econazole concentration due to cyclodextrin complexation, the problem of nasolacrimal drainage remains, along with the issues of unwanted systemic absorption”. but although? 

“Although levels of up to 12.5% cyclodextrin applied to the eye as drops [19, 20], the potential of levels greater than this loaded into soft contact lenses is unknown”. Cyclodextrins is a large family of compounds, please specify. Moreover, 12.5% is very large conc and here in USA, it is only one cyclodextrins approved for ocular delivery and not in such high conc. 

Methods:

“To 1.5 mL of deionized water or PBS (reconstituted tablets) in microcentrifuge tubes was added excess test substance: econazole nitrate, HP-α-CD and HP-β-CD. The tubes were put on a blood tube rotator and left overnight. The following day the tubes were checked for cloudy appearance then centrifuged for 15 min at 3500 RPM, the supernatants were sampled, diluted with water or PBS, and transferred to an HPLC autosampler vials. To determine the effect of cyclodextrin on solubility, a range of concentrations of HP-α-CD and HP-β-CD, up to 13.2% w/v, (actual experiment involved 0.2g in 1.5 mL) was dissolved in PBS solution in microcentrifuge tubes prior to the addition of excess econazole nitrate and then placed in an ultrasonic bath for 30 min. The microcentrifuge tubes were left on the blood tube rotator overnight, then centrifuged for 15 min at 3500 RPM. Supernatants were sampled before being diluted with PBS and transferred to 139

HPLC autosampler vials. Solubility was determined at room temperature and 37 °C”. Poor flow and poor English writing; therefore, the method is confusing.

“The eyeballs were then placed in a water bath at 37 °C”. Why at 37? What is the normal internal temperature of the eye globe?

“Contact lenses were loaded with drug/cyclodextrin using the soaking /breathing in method [15]. These were removed from the packaging and placed in 10 mL of 10% cyclodextrin/econazole nitrate saturated solution overnight in the incubator at 37 °C; up to 12.5% HP-β-CD is considered tolerable [20]. Prior to use, the loaded lenses were removed and gently dried using tissue paper and applied to the porcine corneas, which had been wetted beforehand with 200 μL of PBS”. Poor flow and poor English writing; therefore, the method is confusing.

“After 6 h, the lenses were removed and discarded, and each cornea irrigated with 200 μL of PBS to remove excess drug”. Can you depend on the integrity of the corneal tissues for 6 h in an isolated eye?

Results

Overall, results are not presented well. 

Some examples 

Econazole nitrate was found to have an aqueous solubility of 55.2 ± 4.4 μg/mL at 37ºC - literature values range from <1 mg/mL at 19 ºC and Drug bank. What is drug bank?

Figure 2 and Table 1, shows major dose-dependent increases in econazole solubility in the presence of both HP-α-CD and HP-β-CD (p <0.001). what is dose dependent solubility?

Discussion

Overall, discussion are not presented well and not clear. 

Some examples 

Econazole is a broad-spectrum antifungal in the imidazole drug class, although natamycin tends to be the first-line treatment for fungal keratitis. I do not understand what is the connection?

Precipitation of cyclodextrin in this manner would mean less drug solubilization potential and so less drug in solution, hence loss of linearity (Figure 2). I did not get the point ? do you mean the complex or the cyclodextrin alone?

Author Response

Abstract:

“Hydroxypropyl-α-cyclodextrin (HP-α-CD) and hydroxypropyl-β-cyclodextrin (HP-β-CD) were added incrementally to deionized water or PBS prior to the addition of excess econazole nitrate, and the saturated solubility determined following centrifugation and analysis of the supernatant by reverse phase HPLC”. I think no need for this in the abstract.

This has been amended

“The MIC of econazole was determined against four fungal species associated with keratitis, from which [econazole] required to attain MIC/[econazole] were determined”. What is the solution-only drops ? it is better to use eyedrops.

This was a solution of econazole in PBS

Introduction:

Overall, there is no proper flow and good story in the introduction

Some examples

The introduction part is introducing the disease without any prevalence at least in authors country and without financial burden, and it did not introduce the drug very well. The introduction did not answer the principal question– can we use econazole for ocular delivery? If it is safe, why there is no single approved formulation for ocular applications.

Some of this is covered in the section Clinical Considerations. A sentence has been added to the bottom of P2.

“Natacyn® 5% (natamycin) suspension eye drops and is the first-line topical ophthalmic antifungal preparation in many countries”. 

“The drug has poor penetration into the cornea and the extent of infection can vary widely, therefore duration of treatment can take from a few weeks to months”. It seems unrelated to be included in one sentence.

This has been reworded.

“Increasing the dose of drug in the formulation can result in improved bioavailability and therapeutic outcomes in accordance with Fick’s first law of diffusion, and this should be achievable by changing the drug delivery platform”. This general nonscientific talk and if so needs to be substantiated by references.

The sentence has been amended

“The potential of cyclodextrins to increase the solubility of econazole is not a new concept [9,13,14], but although improved therapeutic outcomes may result from increased econazole concentration due to cyclodextrin complexation, the problem of nasolacrimal drainage remains, along with the issues of unwanted systemic absorption”. but although? 

The sentence has been simplified.

“Although levels of up to 12.5% cyclodextrin applied to the eye as drops [19, 20], the potential of levels greater than this loaded into soft contact lenses is unknown”. Cyclodextrins is a large family of compounds, please specify. Moreover, 12.5% is very large conc and here in USA, it is only one cyclodextrins approved for ocular delivery and not in such high conc. 

Thank you - the error in this sentence has been corrected.

Methods:

“To 1.5 mL of deionized water or PBS (reconstituted tablets) in microcentrifuge tubes was added excess test substance: econazole nitrate, HP-α-CD and HP-β-CD. The tubes were put on a blood tube rotator and left overnight. The following day the tubes were checked for cloudy appearance then centrifuged for 15 min at 3500 RPM, the supernatants were sampled, diluted with water or PBS, and transferred to an HPLC autosampler vials. To determine the effect of cyclodextrin on solubility, a range of concentrations of HP-α-CD and HP-β-CD, up to 13.2% w/v, (actual experiment involved 0.2g in 1.5 mL) was dissolved in PBS solution in microcentrifuge tubes prior to the addition of excess econazole nitrate and then placed in an ultrasonic bath for 30 min. The microcentrifuge tubes were left on the blood tube rotator overnight, then centrifuged for 15 min at 3500 RPM. Supernatants were sampled before being diluted with PBS and transferred to 139

HPLC autosampler vials. Solubility was determined at room temperature and 37 °C”. Poor flow and poor English writing; therefore, the method is confusing.

Thank you - this has all been re-written

“The eyeballs were then placed in a water bath at 37 °C”. Why at 37? What is the normal internal temperature of the eye globe?

This is a very good point, please see added sentence. It is the surface temp of the eye that is most relevant for this work.

“Contact lenses were loaded with drug/cyclodextrin using the soaking /breathing in method [15]. These were removed from the packaging and placed in 10 mL of 10% cyclodextrin/econazole nitrate saturated solution overnight in the incubator at 37 °C; up to 12.5% HP-β-CD is considered tolerable [20]. Prior to use, the loaded lenses were removed and gently dried using tissue paper and applied to the porcine corneas, which had been wetted beforehand with 200 μL of PBS”. Poor flow and poor English writing; therefore, the method is confusing.

This has been revised, and hopefully now flows better.

“After 6 h, the lenses were removed and discarded, and each cornea irrigated with 200 μL of PBS to remove excess drug”. Can you depend on the integrity of the corneal tissues for 6 h in an isolated eye?

Yes!  we have recent data to show that the viability and integrity of the corneal surface for at least 24h. This will be the subject of another paper.

Results

Overall, results are not presented well. 

Some examples 

Econazole nitrate was found to have an aqueous solubility of 55.2 ± 4.4 μg/mL at 37ºC - literature values range from <1 mg/mL at 19 ºC and Drug bank. What is drug bank?

Drug bank is an online resource, reference 28.  The sentence in section 3.1 has been amended.

Figure 2 and Table 1, shows major dose-dependent increases in econazole solubility in the presence of both HP-α-CD and HP-β-CD (p <0.001). what is dose dependent solubility?

This is a scientific term meaning that as the dose of cyclodextrin increases, so does the amount of econazole.   I dont feel any further explanation is warranted in this case.

Discussion

Overall, discussion are not presented well and not clear. 

Some examples 

Econazole is a broad-spectrum antifungal in the imidazole drug class, although natamycin tends to be the first-line treatment for fungal keratitis. I do not understand what is the connection?

This sentence has been amended.

Precipitation of cyclodextrin in this manner would mean less drug solubilization potential and so less drug in solution, hence loss of linearity (Figure 2). I did not get the point ? do you mean the complex or the cyclodextrin alone?

This is an attempt to rationalise the loss of linearity in Fig 2. If some of the cyclodextrin precipitates, then less will be available to solubilise the econazole. The sentence has been amended.

Round 2

Reviewer 1 Report

The authors have addressed the reviewer's comments and I recommend this manuscript be published in Pharmaceutics

Author Response

Thank you

Reviewer 2 Report

I think the current revised version seems OK for me 

Author Response

Thank you

Reviewer 3 Report

The manuscript needs edits in language, methodology, and supporting references for results. Discussion should be elaborated.

Author Response

Some further edits made and 2 references added to the discussion.